geochemistry/materials science/nanotechnology

akaganeite nanorices, sunlight active, mica surfaces, green photocatalyst

**Author for correspondence:**
K. M. Nalin de Silva
e-mail: nalinds@slintec.lk

This article has been edited by the Royal Society of Chemistry, including the commissioning, peer review process and editorial aspects up to the point of acceptance.

# Akaganeite nanorices deposited muscovite mica surfaces as sunlight active green photocatalyst

Anoja Senthilnathan[1,2], D. M. S. N. Dissanayake[2,3], G. T. D. Chandrakumara[3,4], M. M. M. G. P. G. Mantilaka[2,3], R. M. G. Rajapakse[1,5], H. M. T. G. A. Pitawala[3,6] and K. M. Nalin de Silva[2,7]

[1]Academy of Sri Lanka Institute of Nanotechnology (SLINTEC Academy), Mahenwatte, Pitipana, Homagama 10206, Sri Lanka
[2]Sri Lanka Institute of Nanotechnology (SLINTEC), Nanotechnology and Science Park, Mahenwatte, Pitipana, Homagama 10206, Sri Lanka
[3]Postgraduate Institute of Science (PGIS), University of Peradeniya, Peradeniya 20400, Sri Lanka
[4]Department of Science and Technology, Uva Wellassa University, Passara Road, Badulla 90000, Sri Lanka
[5]Department of Chemistry, Faculty of Science, University of Peradeniya, Peradeniya 20400, Sri Lanka
[6]Department of Geology, Faculty of Science, University of Peradeniya, Peradeniya 20400, Sri Lanka
[7]Department of Chemistry, Faculty of Science, University of Colombo, Colombo 00300, Sri Lanka

MMMGPGM, 0000-0001-9832-6807; RMGR, 0000-0003-3943-5362; HMTGAP, 0000-0001-7483-5922; KMNdS, 0000-0003-3219-3233

Thin films of akaganeite [FeO(OH)] nanorices deposited muscovite mica (ANPM) surfaces are synthesized using the facile urea assisted controlled self-assembly technique. The synthesized materials are characterized using scanning electron microscopy (SEM) with energy-dispersive X-ray spectroscopy, atomic force microscopy, X-ray diffraction (XRD), Fourier transform infrared (FT-IR) spectroscopy and thermogravimetric analysis (TGA). The prepared nanorices on mica surfaces show average particle length and width of 200 and 50 nm, respectively. Synthesized material acts as an efficient photocatalyst under UV and sunlight conditions as demonstrated by the degradation of standard methylene blue (MB) solution. The MB degradation efficiencies of the catalyst under exposure to 180 min sunlight and UV are 89% and 87.5%, respectively, which shows that the catalyst is more highly active under sunlight than under UV light. Therefore, the synthesized

material is a potential green photocatalyst in efficient treatment of industrial dye effluents under direct sunlight.

## 1. Introduction

In recent years, photocatalysis has widely been studied and applied in environmental pollution control and prevention due to its ability to break down or convert contaminants into less toxic forms. Most photocatalysts are active under UV light. However, due to adverse health effects and harmfulness towards environment by UV radiation, enormous research and development activities have been carried out in the recent past to synthesize photocatalysts active under visible light [1]. Metal oxide semiconductor materials have been widely studied in photocatalysis [2]. Among the various metal oxide semiconductor photocatalysts, mostly studied $TiO_2$ (3.2 eV) [3,4], $Nb_2O_5$ (3.4 eV) [5], ZnO (3.2 eV) [4] and $WO_3$(2.8 eV) [6] photocatalysts absorb UV light with wavelengths less than 380 nm and visible light with wavelength ranges between 400 and 700 nm, which covers only approximately 5% of the solar spectrum due to their wide band gap [7,8]. These catalysts active under UV radiation show drawbacks in the usage due to their photocorrosion, only active under harmful UV radiation as a result of wide energy band gaps and harmful environmental effects including toxicity. Therefore, such metal oxides and their nanomaterials have been modified to lower bandgap energies in order to synthesize visible light active photocatalysts. Such catalysts are active under direct sunlight, which is very important in designing and running wastewater treatment plants based on photocatalysis with very low cost, as solar energy is readily available. Furthermore, the development of green photocatalysts with features of biocompatibility and none-toxicity is also important when considering the environment and health of species [6]. Therefore, environment-friendly sunlight active efficient green catalysts are much needed timely.

Iron-based nanomaterials [9] have gained a great deal of attraction in recent years as superior photocatalysts, due to their comparatively smaller mean bandgap around 2.2 eV, which cover the wide range of wavelength absorption, which includes both UV and visible regions of solar spectrum [10,11]. Such nanomaterials are also used on several other occasions, including catalyst in Haber process [10], for desulfurization of natural gas, oxidation of alcohol or production of photovoltaic cells for photo-electrochemical hydrogen production and in process of photo-degradation of chlorophenol and azodyes [12,13]. Thin films of iron-based nanomaterials deposited on various surfaces have been applied in most practical applications including the development of sensors of gas, alcohol and humidity [14], as well as in electrodes in lithium-ion batteries [15] and photo-anodes [16]. Iron-based nanomaterials have also been extensively used as synthetic pigments and anticorrosive agents in industries such as paints, ceramics and so on, over the course of recent few decades. When considering the previous research related to the synthesis of the iron nanomaterials based thin films, it is suggested that the materials can be synthesized by methods such as sputtering [17,18], laser ablation [19], electrodeposition [20], spray pyrolysis [21,22], plasma enhanced chemical vapour deposition [23] and aerosol-assisted chemical vapour deposition [23]. Iron-based nanomaterials are effective solution in industrial-scale wastewater treatment by advanced oxidation [24], through which materials provide a promising solution for wastewater treatment due to their production low-cost, strong adsorption capacity, easy separation and enhanced stability [25]. However supportive materials which provide surfaces to attach iron-based nanoparticles are important for the preparation of more effective end-products.

Mica is a layered silicate group of minerals [26,27] which has a characteristic layered lattice comprising two sheets of tetrahedral silica held together by an octahedral alumina sheet with electrostatic forces. Therefore, mica has negative charges on its surface, which gives more importance in the usage of its surface morphology [28]. Mica minerals have outstanding properties including chemical-inertness, dielectric property [26], elasticity, flexibility, resilient property, hydrophilicity [29], insulation, lightweight and optical properties. The special behaviour of mica is that the surface does not get hydroxylated, because the composition of mica is itself hydroxylated and the hydrophilicity of the mica is due to the surface charge of mica. The negatively charged surface of mica is produced as the composition of muscovite mica is $(K,Na)(Al,Mg,Fe)2(Si_3Al_{2.90}H_2KO_{12}Si_{3.10})$, in which the two negatively charged tetrahedral silicate structures are on the surface of the mica. The hydrophilicity of the mica occurs when the $H^+$ ions coming from the water molecules are attracted to negative charges of the surface. Therefore, mica minerals are used as pigments, fillers and insulators for industries such as paint, paper, plastic and electronic industries. However, more studies are required to develop advanced materials from mica minerals for the broad-range

applications due to their exceptional properties. Mica surfaces are capable of providing supportive platform in development of advanced materials by depositing nanoparticles [30] of other materials on mica surfaces. This effort is very important in developing materials for various fields and applications.

In this study, a facile, economical and novel method has been devised for the preparation of the nanometre sized rice-like iron oxide-hydroxide (akaganeite) film on thin mica sheets (referred as ANPM), using urea assisted controlled self-assembly in order to fabricate a sunlight active green photocatalyst, since mica and iron oxides are non-toxic. The soft template of urea is used to synthesize akaganeite nanoparticles (NPs) and deposited using *ex situ* self-assembly of nanoparticles on the surface of the mica. The advantage of this method is modification of the particles and substrates and heating condition are not required to deposit the rice-like morphological akaganeite NPs on the mica substrate. Also, it is a novel effort which is mainly focused on the synthesis of akaganeite NPs and investigation of the sunlight photocatalytic effect of the synthesized nanoparticles on the thin-layer of mica sheet. The coverage and the amount of deposition of the particles were controlled by the concentration of the reactants and time of deposition. The coverage of the nanoparticles, morphological, structural and elemental properties was qualitatively analysed using the observation of scanning electron microscope (SEM) images and energy-dispersive X-ray (EDX) spectroscopy. Thermal stability was ensured using thermogravimetric analysis (TGA). The crystallinity and chemical composition were studied through X-ray Diffraction (XRD). The photocatalytic property of the synthesized material was studied using standard methylene blue (MB) model dye to investigate the application of the material in the treatment of industrial dye effluents. The use of MB is a mere demonstration of the photocatalytic degradation of potential using the synthesized material. Also, MB is a widely used standard dye for studies in photocatalysis. The application part itself makes it a novelty as no previous research has been done based on the property of the synthesized material. The material is photocatalytically active under UV and sunlight. Therefore, the synthesized material is a green photocatalyst in degradation of industrial dye effluents under direct sunlight.

# 2. Material and methods

## 2.1. Materials

Iron(III) chloride (approx. 97% purity), urea (approx. 98% purity) and MB dye (with MW of 373.90) were purchased from Sigma-Aldrich. Mica samples were collected from Matale area which is located at the central part of Sri Lanka.

## 2.2. Preparation of akaganeite NPs

In the synthesis method of akaganeite NPs, 10 ml of 1.11 M urea and 90 ml of 0.37 M iron(III) chloride were mixed together in a two-neck round-bottomed flask while maintaining the temperature at 90°C using reflux heating and stirring for 3 h. The formed precipitate of iron oxide-hydroxide nanoparticles was washed several times and collected by centrifuging and dried at 100°C for 2 h.

## 2.3. Preparation of ANPM

Mica thin layers were cleaved alone the natural cleavage plane (001) with thickness of approximately 200 μm. The sheets were cut into desired square shape (0.5 × 0.5 cm) with a pair of scissors. The cut thin sheets were added to the colloid of akaganeite NPs and were continuously stirred for 24 h. The obtained ANPM was washed with distilled water three times and dried at 100°C.

## 2.4. Characterization of the synthesized material

### 2.4.1. X-ray diffraction

The crystallinity of the raw mica sample and ANPM were analysed using X-ray diffraction patterns from a Siemens D5000 Powder X-ray Diffractometer, with the Cu K$\alpha$ radiation of wavelength $\lambda = 0.1540562$, and the scan rate of $1°$ min$^{-1}$. The obtained XRD patterns were analysed using the X Powder 12 Software with the help of the ICDD PDF2 database. Average crystallite size of the synthesized product was calculated by using the Debye−Scherrer equation which is applied to the major XRD peaks of materials.

### 2.4.2. Morphological analysis

Morphologies of the akaganeite NPs and ANPM was observed using field-emission scanning electron microscope (FE-SEM) Hitachi SU6600. Surface roughness and surface structure were determined by atomic force microscope (AFM) (Park Systems, XE-100) using the cantilever mode (10 nm tip radius) at 0.5 Hz frequency.

### 2.4.3. Elemental analysis

Elemental composition of Fe and O on silica matrix (mica) surface was investigated through EDX spectroscopy with the scanning rate of 192 000 counts s$^{-1}$ for 4.5 min.

### 2.4.4. Thermal analysis

Thermal stability of the synthesized product compared with mica surface with help of thermogravimetric analysis (TGA) (STD Q600) from room temperature to 1000°C at heating rate of 10°C min$^{-1}$ in compressed air medium.

### 2.4.5. Chemical properties

The Fourier transform infrared (FT-IR) spectra of final product, akaganeite nanoparticles and akaganeite nanoparticles deposited mica surface were analysed using Bruker Vertex 80 Fourier transform infrared spectrometer. All spectra were obtained within the range of 500–4000 cm$^{-1}$ with 32 scans per measurement at 0.4 cm$^{-1}$ resolution.

## 2.5. Photocatalytic property of synthesized material

Photo-degradation experiments were carried out in beakers containing 50 ml of MB with various doses (2.0, 2.5, 3.0, 4.0 and 10.0 g) of ANPM photocatalyst. The dye–catalyst mixtures were kept under dark conditions for 30 min before the photo-irradiation in order to obtain the adsorption equilibrium and to deduct the possible error due to the adsorption. After that, samples were exposed to UV-C light (257.5 nm) in a sealed box containing two 36 W, low-pressure mercury vapour discharge lamps (Phillips-TUV 36 W/G36T8). Three millilitre aliquots were taken out from each sample at 15 min time intervals, centrifuged and UV–vis spectra were taken by UV–vis-NIR spectrophotometer (Shimadzu-UV 3600). The same procedure was repeated by exposing the dye solution to sunlight at the 10 g of catalyst dosage in order to use the solar energy source for percentage degradation amount of MB, as calculated by equation (2.1), where $Q_e$ is the percentage degradation of dye, $C_0$ is the initial dye concentration and $C$ is the final dye concentration. The kinetics of each reaction was also determined using equations for first-order kinetics and second-order kinetics is given by the following equations (2.2) and (2.3), respectively.

$$Q_e = \frac{(C_0 - C)}{C_0} \times 100\%, \tag{2.1}$$

$$\ln\frac{C_0}{C} = kt \tag{2.2}$$

and

$$\frac{1}{C} = kt + \frac{1}{C_0}. \tag{2.3}$$

# 3. Results and discussion

## 3.1. XRD characterization of the raw mica, akaganeite NPs and akaganeite NPs – mica materials

Crystallinity and chemical composition of the raw mica material, iron-based nanoparticles and the presence of iron-based nanoparticles on the mica thin film are analysed using XRD patterns. The XRD patterns of the raw material mica confirm the crystallinity of the material through the sharp peaks which are obtained as shown in figure 1a. XRD pattern of raw mica in figure 1 consists of peaks at $2\theta$ values of 8.9°, 17.8°, 26.8°, 35.0°, 45.4° and 55.7° with corresponding basal planes of (003), (006), (009), (112), (0015) and (1114). XRD study reveals that the raw compound consists of the composition as (K,Na)(Al,Mg,Fe)2(Si$_3$Al$_{2.90}$H$_2$KO$_{12}$Si$_{3.10}$). These entire peaks can be assigned to muscovite crystalline form of mica (JCPDS card no. 07-0042). The product IHONPs-mica does not show any considerable changes from muscovite XRD patterns as the lower

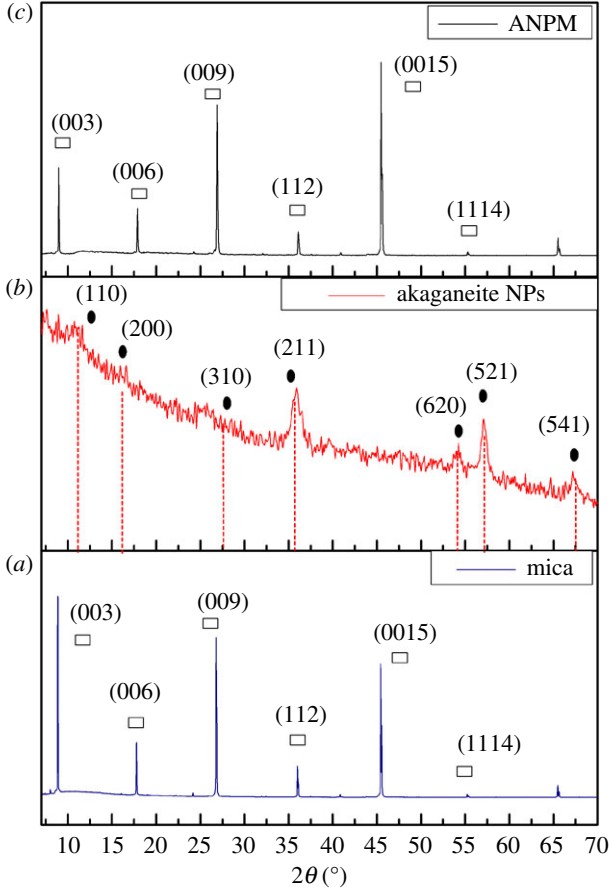

**Figure 1.** XRD patterns of (*a*) mica raw material (*b*) akaganeite NPs (*c*) ANPM.

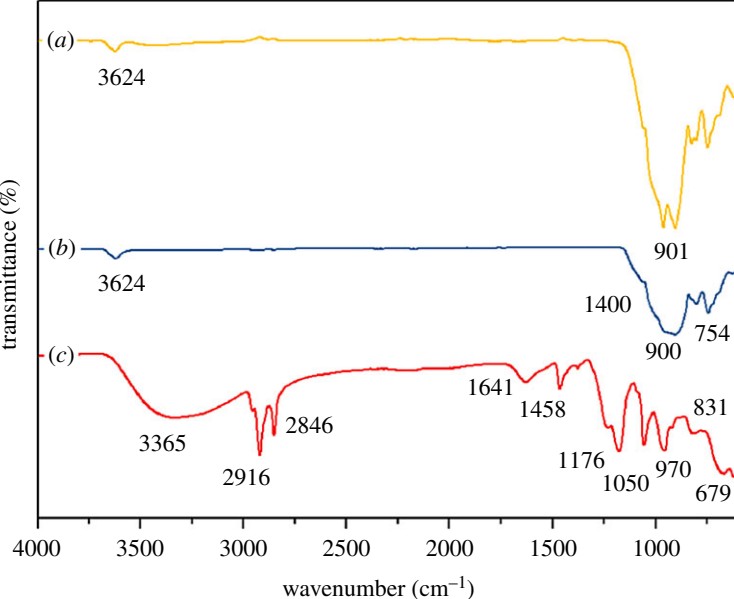

**Figure 2.** The FT-IR spectra of (*a*) ANPM (*b*) mica (*c*) akaganeite NPs.

concentration of akaganeite NPs on mica surface as shown in figure 1*c*. The chemical composition of the synthesized nanoparticles is revealed as iron(III) oxide-hydroxide ($Fe3 + O(OH)$) using the images of the XRD patterns in figure 1*b*, in which, the entire peaks of the pattern can be assigned to crystalline form of akaganeite as the positions and the relative intensities of the peak match to the PDF database2 (JCPDS card no. 42-1315). It consists peaks at 2$\theta$ values of 11.2°, 16.4°, 27.8°, 36.15°, 54.6°, 56.8° and 67.9° with corresponding basal planes of (110), (200), (310), (211), (620), (521) and (541), respectively.

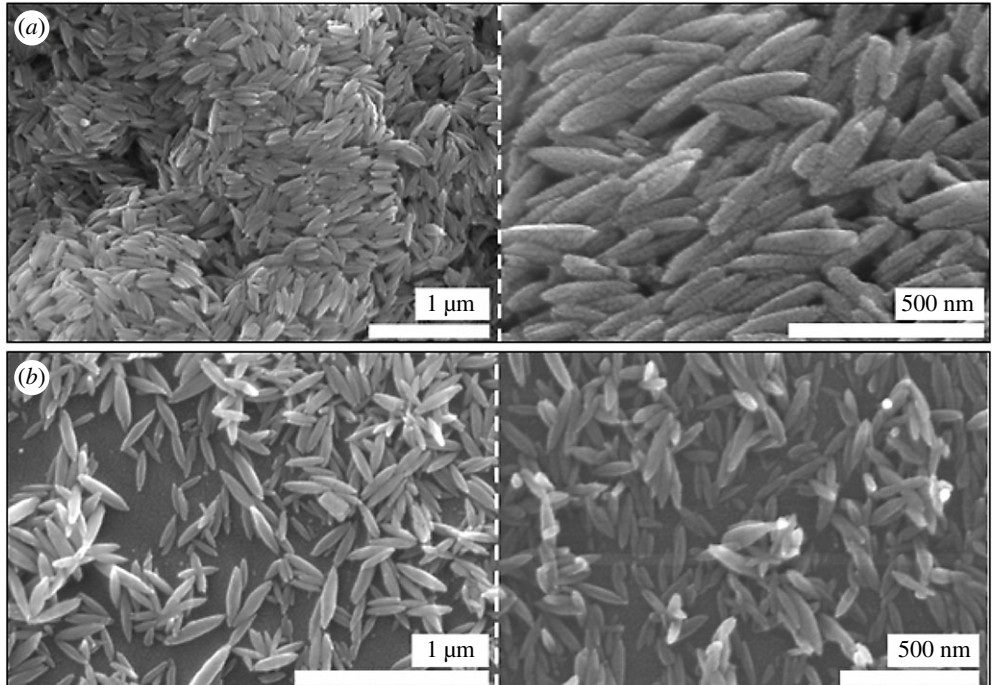

**Figure 3.** (*a*) Synthesized akaganeite NPs at two different magnification, (*b*) synthesized ANPM at two different magnification.

### 3.1.1. FT-IR characterization of the synthesized akaganeite NPs–mica products

FT-IR spectrum illustrated in figure 2*a* confirms the presence of the akaganeite NPs on mica surface due to similar peaks which have been observed from the other spectra in figure 2*b,c*; those are related to the mica raw material and akaganeite NPs, respectively. The bands around 3365 cm$^{-1}$ and 1641 cm$^{-1}$ in figure 2*c* are assigned to the stretching vibration of OH groups and bending vibration of hydroxyl groups or water molecules of the akaganeite NPs, respectively [31,32]. However, the bands at 3365 cm$^{-1}$ and 2916 cm$^{-1}$ ensure that the bands due to bending vibration of hydroxyl groups by the formation of hydroxide groups in the chemical reaction with urea as the product have been analysed without calcination [33]. The akaganeite NPs possess absorbance band in 2916 cm$^{-1}$ due to stretching vibration of C–H bond [34]. In the spectrum of akaganeite NPs the existence of bending vibration of C–O and –NH$_2$ group is attributed to the peak in the region 970 cm$^{-1}$ due to the excessive amount of urea [34]. Characteristic bands for the dissolved carbon dioxide which are evolved gas from the reaction are found at 1458, 1400 and 900 cm$^{-1}$ in figure 2*c*. The band observed at 600 cm$^{-1}$ corresponds to the stretching vibration of the iron metal occupying tetrahedral and octahedral positions. The stretching vibration Fe–O corresponds to tetrahedral iron atoms [35]. Figure 2*a,b* show the similar bands that give the idea of the concentration of the mica material is much higher once it is compared to akaganeite NPs concentration. The characteristic band at about 830–900 cm$^{-1}$ is attributed to the octahedral sheets occupied by a trivalent central atom O–H bending bands, which are assigned to the silicate sheets in the muscovite structure. The weak band at around 3620 cm$^{-1}$ can be corresponded to the OH group, between the tetrahedral and octahedral sheets in the muscovite structure [36]. The bands in 600–750 cm$^{-1}$ are attributed to the bending vibrations of Si–O bond [28]. All the results suggested that the akaganeite NPs have been deposited on the mica surface.

### 3.1.2. Morphological characterization of the synthesized akaganeite NPs, akaganeite NPs–mica products

The FE-SEM images of the akaganeite NPs structure, akaganeite NPs fabricated on mica and elemental composition spectrum at two different magnifications are illustrated in figure 3*a* and *b*, respectively. FE-SEM images reveal the presence of rice-like morphology of akaganeite NPs and ANPM. Akaganeite NPs have an average particle length of 200 nm and average particle width of 50 nm. The dimension of the particles can be clearly seen in the images. These akaganeite NPs are distributed evenly on the surface of the mica. Ninety per cent of the particles show homogeneous structure with similar particle size. However, there are also some irregularities present in this particle size of the rice-like structure.

Surface roughness of the synthesized akaganeite NPs deposited mica is investigated by AFM analysis. The phase and topography images are clearly shown in figure 4*a,b*, respectively. The phase AFM image of

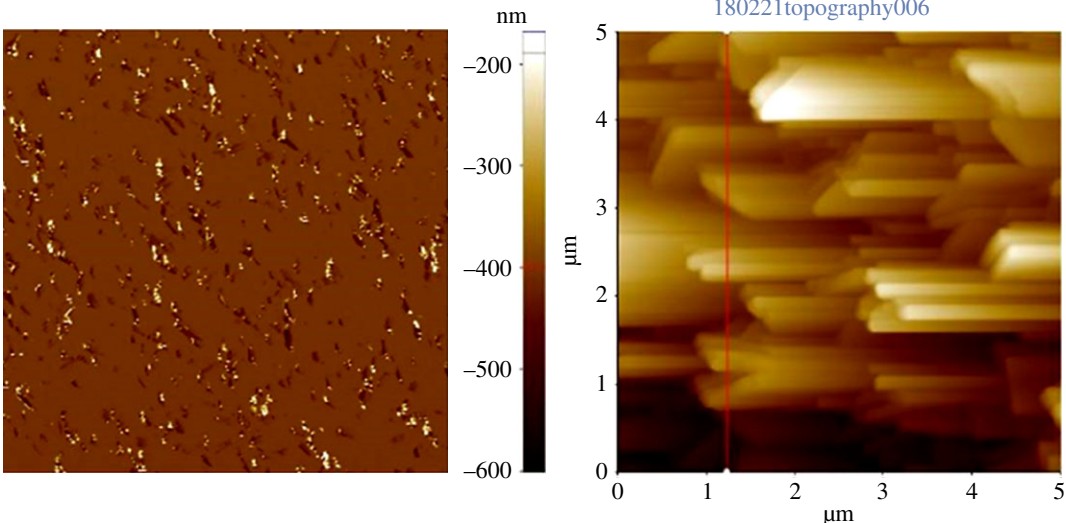

**Figure 4.** AFM images of the prepared ANPM.

the synthesized ANPM is in good agreement with SEM images which show the rice-like morphology. However, the topography images provide slight difference from the SEM images, which shows the rod-shaped morphology may be due to its less accuracy of the topographic image. Surface mean roughness along the red line in figure 4b is 53.64 nm which could be considered as a higher roughness.

### 3.1.3. Elemental analysis

Elemental composition of the synthesized material is determined using EDX to get the qualitative information concerning the elements of the material. As can be seen in figure 5b, the EDX map reveals that synthesized material consists of elements of the muscovite (K, Al, Si, O) and iron. The qualitative information on the weight percentage of the elements in coverage area is also shown. This result also ensures that the 2.7% of iron have distributed on the surface of mica as representative coverage of the material. Elemental analysis of XRF data has been attached as electronic supplementary material, figure S.

### 3.1.4. Thermal characterization of the synthesized ANPM products

Thermal stability of the akaganeite NPs, mica raw material ANPM are illustrated in figure 6. The first mass-loss in the akaganeite NPs curve in figure 6a occurs at around 100°C, which could be due to the removal of absorbed moisture and leftover solvent. The second mass-loss peak at about 120–250°C is found due to thermal decomposition of excess urea, which increases the decomposition due to increasing temperature. The complete decomposition of urea could be finished at the temperature 250°C [37,38]. The third mass-loss is about 5% in the temperature range of 250–500°C, which is due to the thermal decomposition of akaganeite NPs into $Fe_2O_3$ [39]. Thermal decomposition has started at the temperature 250°C and has run up to 500°C in order to form a stable product which is stable up to 1000°C. Mica raw material and ANPM show similar trend of TGA curve, which are illustrated in figure 6b,c. Even though it shows similar behaviour, there is a significant difference in the curve at the temperature in the range from 150°C to 750°C which can be clearly seen in the zoomed image. The synthesized ANPM shows higher stability than that of mica raw material as depicted in the figure. The mass-loss at the temperature range between 150°C and 400°C is considered to be adsorbed water which is different from simple moisture. The mica raw material and ANPM show mass-loss due to physisorbed water at about 0.5% and 0.4%, respectively. The mass-loss at the temperature between 250°C and 500°C may be due to the formation of $Fe_2O_3$ from akaganeite with loss of water content. The mass-loss lowers due to the lower concentration of akaganeite NPs on the surface of the mica. The minimum weight at the temperature 400°C is chosen as dry weight of the ideal muscovite which is equal to the water content between 400°C and 1000°C [40]. The total mass-loss at the temperature 980°C corresponding to muscovite raw material and ANPM are about 4.7%, which is attributed to the loss due to most of the removal of hydroxyl ions below the temperature 850°C as illustrated in the figure [40,41]. Dehydroxylation of muscovite has taken place at wide range of temperature (780–950°C) during dynamic and static heating [40]. Based on TGA curves, it can be

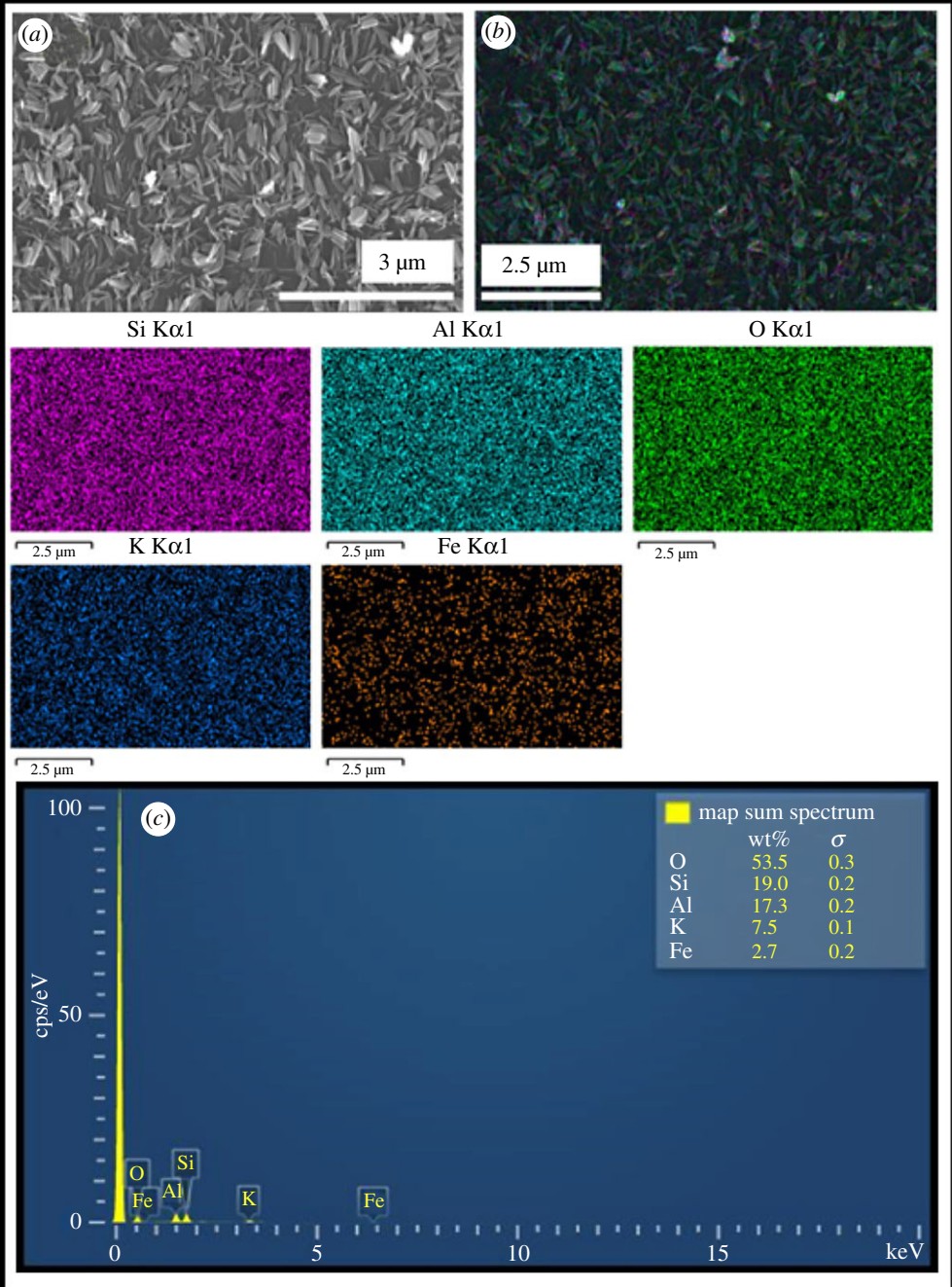

**Figure 5.** EDX mapping of ANPM.

concluded that the synthesized material from akaganeite NPs and mica is thermally stable enough up to the temperature of 1000°C.

### 3.1.5. Formation of the synthesized material ANPM by urea assisted synthesis

At 90°C, urea in the solution is hydrolysed into ammonia which can provide steady and slow release of OH-ions [33] to produce akaganeite NPs with rice-like morphology in the nanometre scale. Mica consists of negative charge on its basal surface as it contains silicate tetrahedral sheets. Thus, the positive electrostatic charge of the akaganeite NPs is attracted to mica surface. Thus, electrostatic force due to electrostatic charge is the driving force which holds the akaganeite NPs on mica. The composite of the material is highly essential in increasing the effectivity of the photocatalytic activity compared to the individual effect of the akaganeite NPs and mica. Thus, interface of the akaganeite NPs and mica is very important in the photocatalytic activity.

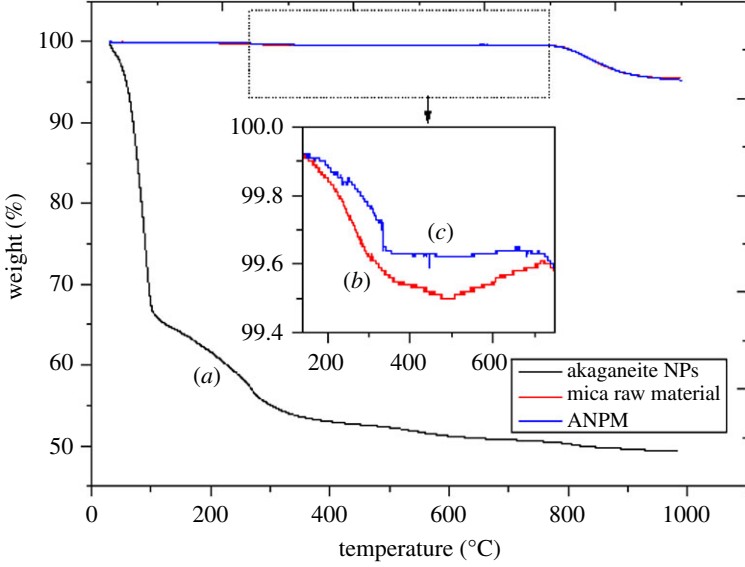

**Figure 6.** TGA plots of (*a*) akaganeite NPs (*b*) mica raw material (*c*) ANPM.

### 3.1.6. Photocatalytic activity of the synthesized material

The effects of catalyst dosage and irradiation time on the percentage degradation of MB aqueous solution have been depicted in figure 7. The maximum absorbance of MB is observed around 663 nm in the UV–vis spectra. The reduction of absorbance as a function of time can be attributed to the photo-degradation of the model organic dye compound by the synthesized material. The photocatalytic experiment is initiated with keeping the dye solution under dark condition for 30 min. The UV–vis spectra do not show any significant variation of the absorbance value after keeping under the dark condition compared to the initial dye solution. The percentage degradation plots indicate the rapid degradation of the aqueous dye while increasing the catalyst dosage. For instance, photocatalytic degradation percentages for 2.0, 2.5, 3.0, 4.0 and 10.0 g of synthesized materials are 46.4%, 57.4%, 65.6%, 71.8% and 87.5% under UV exposure within 180 min. About 89.04% of the dye solution has been degraded by 10.0 g ($\pm$0.0001) of ANPM within 180 min under exposure of sunlight. The degradation efficiency of the novel material is higher under the sunlight than that of the UV-C light. There were no significant variations of the UV–vis spectra observed when the dye solutions were exposed to the UV light and sunlight without ANPM material.

Mica sheets provide a suitable substrate to deposit akaganeite NPs from the precursor solution. The catalytic activity of the material is expected to be enhanced due to the high surface area of the material and properties of the surface. This can be further explained by the terms of high concentration of active sites on the surface and high organic dye molecules absorbed on to the nanometre scale catalytic surface [42].

The electrons in the valence band (VB) of ANPM have been excited to the conduction band by leaving holes in the VB. Photo-induced electrons can react with oxygen absorbed to the material surface or oxygen dissolved in the water. This reduction process creates super oxide radicals ($\bullet O_2$) which can participate in the degradation process. The adsorbed water molecules on surfaces of akaganeite NPs and mica play a major role in the formation of hydroxyl radicals ($\bullet OH$) through the reaction with photo-generated holes or superoxide radicals at the surface of the photocatalyst [43]. Furthermore, these oxygen free radicals react with $H^+$ and produce hydroperoxyl radicals ($\bullet OOH$) and $H_2O_2$. Moreover, further reduction mechanism can occur by providing ($\bullet OH$) [24,42,43]. These free radicals formed during the process degrade the model organic dye which is a pollutant and release $CO_2$ and $H_2O$ as non-harmful products. The photo-degradation of MB follows the following reactions and steps of mechanism:

1. Absorption of required photon energy by the photocatalyst ANPM

$$\text{ANPM} + h\nu \longrightarrow e_{CB}^- + h_{VB}^+.$$

2. Superoxide formation through oxygen reduction reaction

$$O_2 + e_{CB}^- \longrightarrow O_2^{\bullet -}.$$

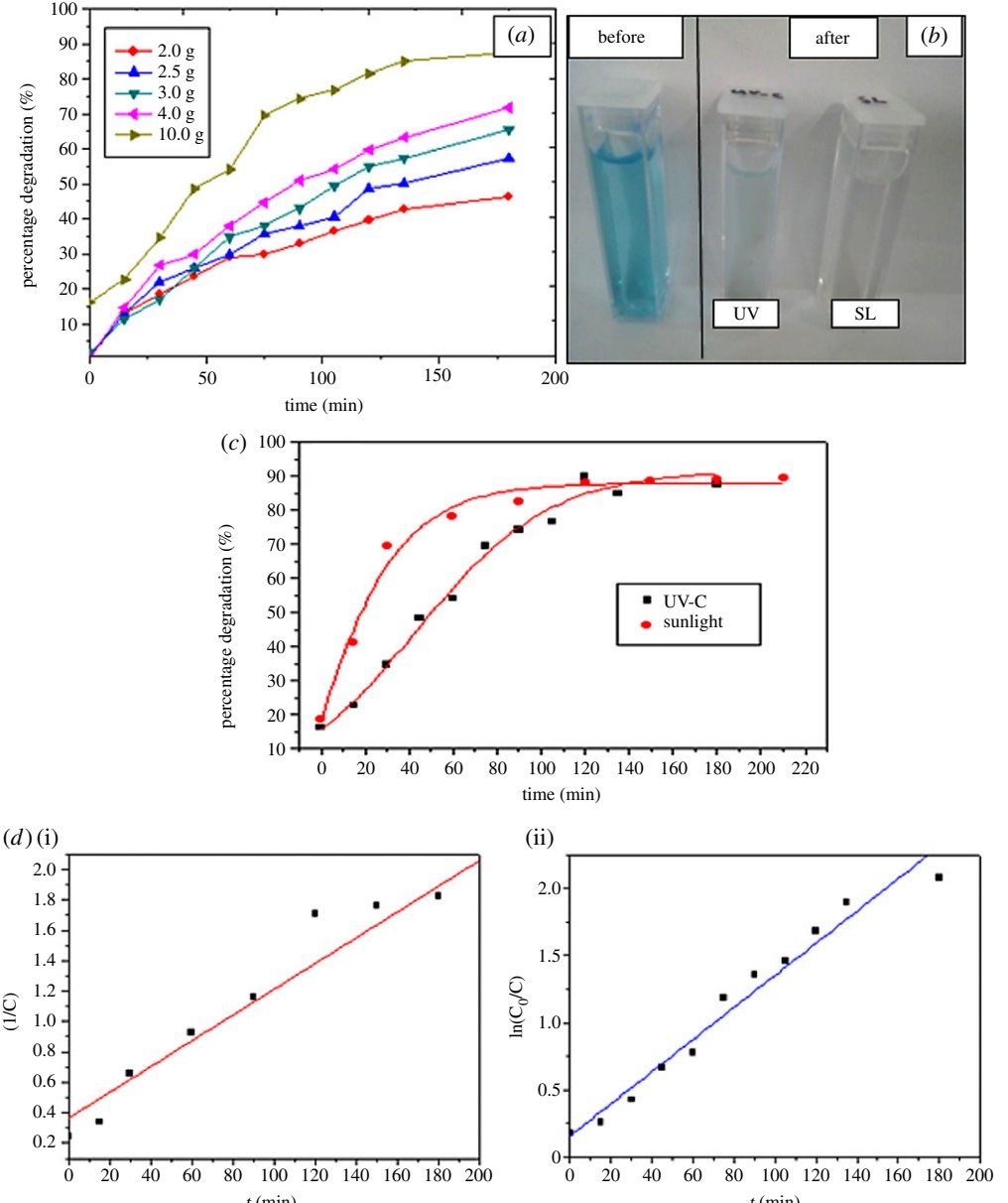

**Figure 7.** (a) Percentage degradation of 5 ppm MB dye with different catalyst dosage (ANPM) and time, (b) photographs of 5 ppm MB dye solution before and after irradiation with UV-C light and sunlight in the presence of ANPM, (c) percentage degradation of 5 ppm MB dye degradation by 10 g ANPM with time under UV-C light and sunlight, (d) degradation kinetics (i) UV-C light (ii) sunlight.

3. Highly reactive $OH^{\bullet}$ formation from water molecules

$$(H_2O \leftrightarrow H^+ + OH^-) + h_{VB}^+ \longrightarrow H^+ + OH^{\bullet}.$$

4. Neutralization of superoxide

$$O_2^{\bullet -} + H^+ \longrightarrow HO_2^{\bullet}.$$

5. Formation of hydrogen peroxide

$$2HO_2^{\bullet} \longrightarrow H_2O_2 + O_2.$$

6. Formation of highly reactive $OH^{\bullet}$ radicals

$$H_2O_2 + e^- \longrightarrow OH^{\bullet} + OH^-.$$

7. Degrading of the MB

$$R \text{ (methylene blue)} + OH^{\bullet} \longrightarrow R^{\bullet} + H_2O$$

$$R + h^+(\text{hole}) \longrightarrow R^{+\bullet} \longrightarrow \text{degradation Products.}$$

8. End reaction [44,45]

$$RCOO^- + h^+(\text{hole}) \longrightarrow R^{\bullet} + CO_2.$$

Therefore, ANPM can be used as an alternative for conventional photocatalyst in the treatment of industrial dye effluents.

The plots of $\ln(C_0/C)$ versus irradiation time under UV-C light and $(1/C)$ versus irradiation time under sunlight are shown in figure 7d (i) and (ii), respectively, in order to indicate kinetics of the reaction. This confirms that the MB degradation by synthesized material follows first-order kinetics under sunlight and second-order kinetics under sunlight. The rate constant (k), half-life ($t_{1/2}$) and linear coefficient ($R^2$) under UV-C light are $0.0119 \text{ min}^{-1}$, 58.25 min and 0.96606, respectively. The same parameters under sunlight are $0.0084 \text{ min}^{-1}$, 23.80 min and 0.92287, respectively. It can be concluded with the data of MB dye percentage degradation that photocatalyst effectively degrade the dye with short period of time under sunlight.

# 4. Conclusion

The rice-like morphological akaganeite NPs on mica have been synthesized using urea assisted controlled self-assembly where the controlled parameters are deposition time and the concentration of the reactants. XRD study reveals that the raw compound consists of the composition $(K,Na)(Al,Mg,Fe)_2(Si_3Al_{2.90}H_2KO_{12}Si_{3.10})$, which can be assigned the material is the muscovite form of mica. According to the SEM, average length and width of akaganeite NPs which have been employed to deposit on mica surface are 200 and 50 nm, respectively. All the results of FT-IR proved that akaganeite NPs have been deposited on the mica surface with its chemical analysis. From TGA curves it can be concluded that the synthesized material from akaganeite NPs and mica are thermally stable enough up to the temperature of $1000^\circ C$. Synthesized ANPM showed superior photocatalytic degradation of MB dye under sunlight. Therefore, ANPM is the alternative material for photocatalysts that can be used for treatment of industrial dye effluents.

Data accessibility. The datasets supporting this article have been uploaded as part of the electronic supplementary material. Data available from the Dryad Digital Repository: https://doi.org/10.5061/dryad.mv3276v [46].
Authors' contributions. M.M.M.G.P.G.M. conceived the idea and designed the project. A.S., D.M.S.N.D. and G.T.D.C. carried out the experiments with different contributions and wrote the first draft of the paper. H.M.T.G.A.P. contributed towards expert guidance of materials and their characteristics. R.M.G.R. and K.M.N.D.S. supervised the project and helped to fine-tune the manuscript. All authors discussed the results and commented on the final manuscript.
Competing interests. We declare we have no competing interests.
Funding. Financial assistance by the National Research Council, Sri Lanka, grant no. 16–123 for providing funds for chemicals and consumables is highly acknowledged.

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
