## [Reviewer comments · Royal Society Open Science]

Review History

RSOS-182212.R0 (Original submission)

Review form: Reviewer 1

Is the manuscript scientifically sound in its present form?

Yes

Are the interpretations and conclusions justified by the results?

Yes

Is the language acceptable?

Yes

Is it clear how to access all supporting data?

Yes

Do you have any ethical concerns with this paper?

No

Have you any concerns about statistical analyses in this paper?

No

Recommendation?

Accept with minor revision (please list in comments)

Comments to the Author(s)

1. Why did authors prefer to use "Akaganeite" instead of " $\text{Fe}_3\text{O}(\text{OH})$ " since the XRD pattern shows completely $\text{Fe}_3\text{O}(\text{OH})$?
2. It was mentioned that mica is chemically inert but at the same place it says it is hydrophilic. Does surface get hydroxylated?
3. Why did the authors choose methylene blue that has been well studied already?
4. What was the thickness of mica thin sheets? How was it cut?
5. How is the interface of mica and akaganeite? Is it important for photocatalytic activity?
6. The authors did not discuss the important role of water molecules in akaganeite and mica on the photocatalytic activity. Please see the following references: Applied Clay Science 101 (2014) 38–43;
7. Adsorption of organic molecules on the surface of photocatalyst is important parameter, and it needs to be addressed.
8. Various pathways for the photodegradation of methylene blue has been reported and the authors need to identify which reaction pathway took place here.

Review form: Reviewer 2

Is the manuscript scientifically sound in its present form?

Yes

Are the interpretations and conclusions justified by the results?

Yes

Is the language acceptable?

Yes

Is it clear how to access all supporting data?

Yes

Do you have any ethical concerns with this paper?

No

Have you any concerns about statistical analyses in this paper?

No

Recommendation?

Accept with minor revision (please list in comments)

Comments to the Author(s)

This is a neat article which should be accepted provided the minor typos are corrected through a revision.

Decision letter (RSOS-182212.R0)

08-Feb-2019

Dear Professor Nalin De Silva:

Title: Akaganeite nanorices deposited muscovite mica surfaces as sunlight active green photocatalyst
Manuscript ID: RSOS-182212

Thank you for submitting the above manuscript to Royal Society Open Science. On behalf of the Editors and the Royal Society of Chemistry, I am pleased to inform you that your manuscript will be accepted for publication in Royal Society Open Science subject to minor revision in accordance with the referee suggestions. Please find the reviewers' comments at the end of this email.

The reviewers and handling editors have recommended publication, but also suggest some minor revisions to your manuscript. Therefore, I invite you to respond to the comments and revise your manuscript.

Please also include the following statements alongside the other end statements. As we cannot publish your manuscript without these end statements included, if you feel that a given heading is not relevant to your paper, please nevertheless include the heading and explicitly state that it is not relevant to your work. We have included a screenshot example of the end statements for reference.

- Ethics statement

Please clarify whether you received ethical approval from a local ethics committee to carry out your study. If so please include details of this, including the name of the committee that gave consent in a Research Ethics section after your main text. Please also clarify whether you received informed consent for the participants to participate in the study and state this in your Research Ethics section.

OR

Please clarify whether you obtained the necessary licences and approvals from your institutional animal ethics committee before conducting your research. Please provide details of these licences and approvals in an Animal Ethics section after your main text.

OR

Please clarify whether you obtained the appropriate permissions and licences to conduct the fieldwork detailed in your study. Please provide details of these in your methods section.

- Funding statement

Please include a funding section after your main text which lists the source of funding for each author.

Because the schedule for publication is very tight, it is a condition of publication that you submit the revised version of your manuscript before 17-Feb-2019. Please note that the revision deadline will expire at 00.00am on this date. If you do not think you will be able to meet this date please let me know immediately.

Best wishes,
Dr Laura Smith
Publishing Editor, Journals

Royal Society of Chemistry
Thomas Graham House

Science Park, Milton Road
Cambridge, CB4 0WF
Royal Society Open Science - Chemistry Editorial Office

On behalf of the Subject Editor Professor Anthony Stace and the Associate Editor Professor Tobias Hertel.

RSC Associate Editor:
Comments to the Author:
(There are no comments.)

RSC Subject Editor:
Comments to the Author:
(There are no comments.)

Reviewer comments to Author:
Reviewer: 1

Comments to the Author(s)

1. Why did authors prefer to use "Akaganeite" instead of "Fe₃O(OH)" since the XRD pattern shows completely Fe₃O(OH)?
2. It was mentioned that mica is chemically inert but at the same place it says it is hydrophilic. Does surface get hydroxylated?
3. Why did the authors choose methylene blue that has been well studied already?
4. What was the thickness of mica thin sheets? How was it cut?
5. How is the interface of mica and akaganeite? Is it important for photocatalytic activity?
6. The authors did not discuss the important role of water molecules in akaganeite and mica on the photocatalytic activity. Please see the following references: Applied Clay Science 101 (2014) 38–43;
7. Adsorption of organic molecules on the surface of photocatalyst is important parameter, and it needs to be addressed.
8. Various pathways for the photodegradation of methylene blue has been reported and the authors need to identify which reaction pathway took place here.

Reviewer: 2

Comments to the Author(s)

this is a neat article which should be accepted provided the minor typos are corrected through a revision.

Author's Response to Decision Letter for (RSOS-182212.R0)

See Appendix A.

Decision letter (RSOS-182212.R1)

25-Feb-2019

Dear Professor Nalin De Silva:

Title: Akaganeite nanorices deposited muscovite mica surfaces as sunlight active green photocatalyst
Manuscript ID: RSOS-182212.R1

It is a pleasure to accept your manuscript in its current form for publication in Royal Society Open Science. The chemistry content of Royal Society Open Science is published in collaboration with the Royal Society of Chemistry.

On behalf of the Subject Editor Professor Anthony Stace and the Associate Editor Professor Tobias Hertel.

RSC Associate Editor
Comments to the Author:
(There are no comments.)

Reviewer(s)' Comments to Author:

Appendix A

Responses to Reviewers' comments and Suggestions

Title of Initial Submission: Akaganeite nanorices deposited muscovite mica surfaces as sunlight active green photocatalyst

Manuscript ID: RSOS-182212

Responses to Reviewer No. 1

We thank reviewer for careful evaluation of our manuscript and deciding to accept with some minor corrections. These comments are very useful to improve our manuscript. Below are responses to your comments and we made changes accordingly. Changes made in text are highlighted in blue colour font.

Reviewer's Comment: Why did authors prefer to use "Akaganeite" instead of "Fe₃O(OH)" since the XRD pattern shows completely Fe₃O(OH)?

Our Response: Thanks for showing this problem of our manuscript. The XRD analysis suggest that the crystalline form of the Fe₃O(OH) nanoparticles is akaganeite (JCPDS card no. 42-1315). Therefore we preferred to use name of the compound. We have modified the XRD figure accordingly and made required changes in the text.

Reviewer's Comment: It was mentioned that mica is chemically inert but at the same place it says it is hydrophilic. Does surface get hydroxylated?

Our Response: We thank the reviewer for raising this question. The surface of mica does not get hydroxylated. Because, the composition of mica is itself hydroxylated and the hydrophilicity of the mica is due to the surface charges of mica. The negatively charged surface of mica is produced as the composition

of muscovite mica is $(K,Na)(Al,Mg,Fe)_2(Si_3Al_{2.90}H_2KO_{12}Si_{3.10})$ in which, the two negatively charged tetrahedral silicate structures are on the surface of the mica. The hydrophilicity of the mica occurred when the H^+ ions coming from water molecule is attracted to negative charge of the surface. This point is now included in the reviewed manuscript.

Reviewer's Comment: Why did the authors choose methylene blue that has been well studied already?

Our Response: We thank reviewer for comment. The main focus of the research is not studying the photocatalytic degradation of methylene blue. The use of methylene blue is a mere demonstration of the photocatalytic degradation potential of the synthesized material. Methylene blue is the standard dye (ASTM) to study the photocatalysis.

Reviewer's Comment: What was the thickness of mica thin sheets? How was it cut?

Our Response: We thank reviewer for highlighting questions to improve the manuscript. We have modified the Experimental section as "Mica thin-layers were cleaved along the natural cleavage plane (001) with thickness approximately 200 μm . The sheets were cut into desired square shape (0.5 cm \times 0.5 cm) by a pair of scissors".

Reviewer's Comment: How is the interface of mica and akaganeite? Is it important for photocatalytic activity?

Our Response: We thank reviewer for comment. Mica consists negative charges on its basal surface as it contains silicate tetrahedral sheets. Thus, the positive electrostatic charge of the akaganeite NPs is attracted to mica surface. Thus, electrostatic force due to electrostatic charge is the driving force which

holds the akaganeite NPs on mica. The composite of the material is highly increasing the effectively of the photocatalytic activity compared to the individual effect of the akaganeite NPs and mica. Thus, interface of the akaganeite NPs and Mica is very important in the photocatalytic activity. We have now mentioned this in discussion section of revised manuscript.

Reviewer's Comment: The authors did not discuss the important role of water molecules in akaganeite and mica on the photocatalytic activity. Please see the following references: Applied Clay Science 101 (2014) 38–43.

Our Response: We thank reviewer for suggestion to improve manuscript. We have cited this article in the revised manuscript. We have included regarding the role of water molecules in the photocatalytic activity in the revised manuscript as follows: “The adsorbed water molecules on surfaces of akaganeite NPs and mica play a major role in formation of hydroxyl radicals (.OH) through the reaction with photo-generated holes or superoxide radicals at the surface of the photocatalyst [43]”.

Reviewer's Comment: Adsorption of organic molecules on the surface of photocatalyst is important parameter, and it needs to be addressed.

Our Response: We thank reviewer for stating drawbacks of the manuscript to improve it. The description of the adsorption of organic molecules on the surface of photo-catalyst is included in the section of results and discussions of revised manuscript as follows: “The photocatalytic experiment is initiated with keeping the dye solution under dark condition for 30 min. The UV-Vis spectra do not show any significant variation of the absorbance value after keeping under the dark condition compared to the initial dye solution”.

Reviewer's Comment: Various pathways for the photodegradation of methylene blue have been reported and the authors need to identify which reaction pathway took place here.

Our Response: We thank reviewer for suggestion. We have included the pathway for the photo-degradation of methylene blue in the section of Results and Discussions as follows: "The photo-gradation of methylene blue follows the following steps of mechanism:

1. Absorption of required photon energy by the photo-catalyst ANPM

2. Superoxide formation through oxygen reduction reaction

3. Highly reactive OH[•] formation from water molecules

4. Neutralization of superoxide

5. Formation of Hydrogen peroxide

6. Formation of highly reactive OH[•] radicals

7. Degrading of the methylene blue

8. End Reaction

[2]"

Responses to Reviewer No. 2

Reviewer's Overview: This is a neat article which should be accepted provided the minor typos are corrected through a revision.

Our Response: We thank reviewer for careful evaluation of our manuscript and recommending to accept after minor corrections. We have done required changes and revised the manuscript accordingly.